# Relationships between Heart Chamber Morphology or Function and Respiratory Parameters in Patients with HFrEF and Various Types of Sleep-Disordered Breathing

**DOI:** 10.3390/diagnostics13213309

**Published:** 2023-10-25

**Authors:** Karolina Simionescu, Danuta Łoboda, Mariusz Adamek, Jacek Wilczek, Michał Gibiński, Rafał Gardas, Jolanta Biernat, Krzysztof S. Gołba

**Affiliations:** 1Department of Electrocardiology and Heart Failure, Medical University of Silesia, 40-635 Katowice, Poland; ksimionescu@sum.edu.pl (K.S.); jwilczek@sum.edu.pl (J.W.); mgibinski@sum.edu.pl (M.G.); rafal.gardas@sum.edu.pl (R.G.); kgolba@sum.edu.pl (K.S.G.); 2Department of Electrocardiology, Upper-Silesian Medical Centre, 40-635 Katowice, Poland; 3Department of Thoracic Surgery, Medical University of Silesia, 40-055 Katowice, Poland; madamek@sum.edu.pl; 42nd Department of Radiology, Medical University of Gdansk, 80-210 Gdansk, Poland

**Keywords:** heart failure with reduced ejection fraction, central sleep apnea, obstructive sleep apnea, sleep-disordered breathing

## Abstract

Sleep-disordered breathing (SDB), i.e., central sleep apnea (CSA) and obstructive sleep apnea (OSA), affects the prognosis of patients with heart failure with reduced ejection fraction (HFrEF). The study assessed the relationships between heart chamber size or function and respiratory parameters in patients with HFrEF and various types of SDB. The 84 participants were patients aged 68.3 ± 8.4 years (80% men) with an average left ventricular ejection fraction (LVEF) of 25.5 ± 6.85% who qualified for cardioverter-defibrillator implantation with or without cardiac resynchronization therapy. SDB, defined by an apnea–hypopnea index (AHI) ≥ five events/hour, was diagnosed in 76 patients (90.5%); SDB was severe in 31 (36.9%), moderate in 26 (31.0%), and mild in 19 (22.6%). CSA was the most common type of SDB (64 patients, 76.2%). A direct proportional relationship existed only in the CSA group between LVEF or stroke volume (SV) and AHI (*p* = 0.02 and *p* = 0.07), and between LVEF or SV and the percentage of total sleep time spent with hemoglobin oxygen saturation < 90% (*p* = 0.06 and *p* = 0.07). In contrast, the OSA group was the only group in which right ventricle size showed a positive relationship with AHI (for basal linear dimension [RVD1] *p* = 0.06), mean duration of the respiratory event (for RVD1 *p* = 0.03, for proximal outflow diameter [RVOT proximal] *p* = 0.009), and maximum duration of respiratory event (for RVD1 *p* = 0.049, for RVOT proximal *p* = 0.006). We concluded that in HFrEF patients, SDB severity is related to LV systolic function and SV only in CSA, whereas RV size correlates primarily with apnea/hypopnea episode duration in OSA.

## 1. Introduction

Sleep-disordered breathing (SDB), which includes central sleep apnea (CSA) and obstructive sleep apnea (OSA), is common in patients with heart failure with reduced ejection fraction (HFrEF) [1,2,3,4,5].

The characteristic feature of OSA is the recurrent loss of upper airway patency resulting from the collapse of the pharyngeal tissues while maintaining respiratory muscle function. Obesity, age, large neck circumference, male sex, and family history are risk factors for OSA [6,7]. On the other hand, OSA is a significant risk factor for hypertension [8,9], arrhythmias and conduction disorders [10], coronary artery disease [11], pulmonary hypertension, stroke [12], heart failure (HF) [13,14], or sudden cardiac death (SCD) [15].

CSA is characterized by recurrent apneic episodes accompanied by a lack of respiratory effort. In a large prospective study involving 2865 participants [16], CSA was a predictor of HF decompensation. That suggests that CSA is not simply a marker of severe HF but may also precede the onset of symptomatic HF in patients with subclinical ventricular dysfunction [17]. In HFrEF patients, the nocturnal rostral fluid shifting from the lower extremities and an increased neck circumference contribute to the pathogenesis of OSA [18,19]. On the other hand, the increased left ventricular (LV) filling pressure engenders pulmonary congestion, leading to the activation of lung receptors that stimulate hyperventilation. Therefore, the partial pressure of carbon dioxide (PCO_2_) falls below the threshold level required to stimulate breathing, triggering CSA [18,20,21]. In those cases, CSA is most often a Cheyne–Stokes Respiration (CSR) pattern component. The CSA–CSR is a form of periodic breathing in which central apnea alternates with periods of hyperventilation, and tidal volume increases and decreases in crescendo–decrescendo mode [22].

The coexistence of CSA and OSA in HFrEF patients is common, and the central-to-obstructive-events ratio in the same patient may vary overnight (depending on sleep phase) and in longer follow-up (depending on fluid overload and the presence of pulmonary congestion) [4,23,24]. Repetitive apneas/hypopneas and compensatory hyperpnea in OSA and CSA are associated with adverse cardiovascular consequences. The most important are periodic hypoxemia–reoxygenation with fluctuations in pCO_2_, frequent awakenings with sleep architecture disorders, increased sympathetic activity, and fluctuations in the negative intrathoracic pressure [17,25,26]. Nevertheless, the differences between the pathomechanisms of CSA and OSA may lead to various cardiovascular changes, depending on the dominant type of sleep apnea. That may be of importance in the diagnosis, and patient management.

This study assessed the relationships between heart chamber size or function and respiratory parameters in patients with HFrEF and various types of SDB.

## 2. Materials and Methods

### 2.1. Study Group

The study group consisted of patients with HFrEF referred to the Electrocardiology Department of the Upper-Silesian Medical Center (Katowice, Poland) for a cardioverter-defibrillator (ICD) implant, with or without cardiac resynchronization therapy (CRT) in primary prevention of SCD. The study was part of the research concerning the predicting factors of the response to electrotherapy carried out in our center. Optimal pharmacotherapy (OMT) for at least three months prior to inclusion and the absence of circulatory decompensation symptoms were definite criteria. None of the patients had been tested for SDB. Prior to device implantation, all subjects underwent physical examination involving anthropometric measurements, electrocardiogram, two-dimensional and Doppler echocardiography, as well as a six-minute walk test (6 MWT) and polysomnography type I. All subjects completed the Epworth Sleepiness Scale (ESS) [27] to assess the severity of daytime sleepiness.

The Bioethics Committee of the Silesian Medical University in Katowice (resolution No. KNW/0022/KB1/17/15/16/17 dated 10 January 2017) expressed a positive opinion on conducting the study.

### 2.2. Echocardiographic Measurements

All patients underwent complete transthoracic echocardiography with an iE33 ultra-sound (Phillips, Amsterdam, The Netherlands), according to the American Society of Echocardiography and the European Association of Cardiovascular Imaging recommendation [28]. We measured LV end-diastolic volume (LVEDV) index, LV end-systolic volume (LVESV) index, and LV stroke volume (SV) calculated as the difference between LVEDV and LVESV. LV ejection fraction (LVEF) was determined using a biplane disk summation method (modified Simpson’s rule). We assessed the right ventricular (RV) basal linear dimension (RVD1) and proximal outflow diameter (RVOT proximal). We determined RV systolic longitudinal function by a tricuspid annular systolic excursion (TAPSE). We calculated RV systolic pressure (RVSP) from a peak tricuspid systolic pressure gradient (TRPG) and estimated right atrial pressure.

### 2.3. Polysomnographic Measurements

Full polysomnography (PSG) using Alice 6 Diagnostic Sleep System (Philips Respironics, Amsterdam, The Netherlands) was obtained in all patients in the study. PSG was analyzed according to the American Academy of Sleep Medicine [29]. Apnea was defined as a reduction in nasal airflow by >90% for >10 s. Hypopnea was defined as a reduction in airflow by ≥50% for at least 10 s with a drop in hemoglobin oxygen saturation (SpO_2_) ≥ 3% from the preceding stable saturation or an arousal. SDB was analyzed only in terms of CSA vs. OSA. In addition to assessing the apnea–hypopnea index (AHI) to determine the number of apnea and hypopnea events per hour of sleep (events/h), we analyzed the average (mean A/H) and maximum (max A/H) duration of the respiratory event, the average (mean SpO_2_) and minimum (min SpO_2_) SpO_2_ during sleep, and the percentage of sleep time with SpO_2_ below 90% (TST90), calculated as the ratio of sleep time with hemoglobin saturation <90% to total sleep time ×100%.

### 2.4. Statistical Analysis

Statistical analysis was performed using MedCalc version 20.014 software (MedCalc Software Ltd., Ostend, Belgium). Qualitative data were expressed in numbers and percentages and compared using the chi-square test. The quantitative parameters were presented as the mean (X) and standard deviation (SD), or the median (M) and interquartile range (IQR). The normality distribution was assessed with the Shapiro–Wilk test. The homogeneity of variance in the compared groups was tested by the F-test or Levene’s test. Univariate analysis was performed by: (1) Assessing the significance of differences for two independent samples of variables with normal distribution using students’ *t*-tests (with or without Welch’s correction depending on the result of the F-test) or using the Mann–Whitney U test if the distribution of variables was different from normality, (2) Linear regression analysis. A two-tailed *p*-value of <0.05 was considered statistically significant.

## 3. Results

The study included 84 patients with a mean LVEF of 25.5 ± 6.85% eligible for high voltage therapy according to the Class I or IIa criteria of the ESC guidelines for pacing and resynchronization therapy [30] and HF [31]. Finally, a CRT-defibrillator (CRT-D) was implanted in 59 patients (71.1%), a CRT-pacemaker (CRT-P) in 1 patient, and an ICD in 17 patients. The remaining seven patients qualified for further pharmacological treatment.

Men constitute 80% (67 patients), and women 20% (17 patients) of the study group. The mean age of the participants was 68.3 ± 8.4 years, the mean BMI was 30.4 ± 6.3 kg/m^2^, and the mean neck circumference was 42 ± 3.9 cm.

Sleep apnea, defined by an AHI value ≥5, was diagnosed in 76 patients (90.5%), with severe apnea (AHI > 30 events/h) occurring in 31 patients (36.9%), moderate apnea (AHI 15–30 events/h) in 26 patients (31.0%), and mild apnea (AHI 5–14 events/h) in 19 patients (22.6%). CSA was the most common form of SDB and occurred in 64 patients (76.2%) (CSA group), whereas obstructive incident apnea was the predominant form in the remaining 12 patients (14.3%) (OSA group). In the group of patients, eight (9.5%) were not found to have SDB.

There were no sex- or age-related differences between the CSA and OSA groups (Table 1). The percentage distribution of patients with CSA and OSA did not differ between the groups of men and women (*p* = 0.93). However, the number of women in the OSA group was low (two patients). The OSA group had a higher BMI than the CSA group (*p* = 0.001). Atrial fibrillation (AF) was significantly more common in the CSA group than in the OSA group (*p* = 0.03). Pharmacological treatment was similar in both groups (Table 1). Due to the lack of reimbursement by the Polish insurer, none of the patients received treatment with sacubitril/valsartan or type 2 sodium–glucose co-transporters.

There were no differences in HFrEF etiology and HFrEF duration between the CSA and OSA groups (*p* = 0.71 and *p* = 0.10, respectively) (Table 2). There were also no differences between groups in parameters assessing cardiorespiratory fitness (CRF), i.e., NYHA and 6 MWT. In echocardiographic measurements, OSA patients showed a trend toward thicker IVS (*p* = 0.06) and higher TAPSE (*p* = 0.07).

In addition, there was a trend toward higher ESS scores in the OSA vs. CSA group (*p* = 0.08). The vast majority of patients had both central and obstructive apneas. Only in five participants was the rate of central apneas 100%, and in four participants the rate of obstructive apneas was 100%. There were no differences between the CSA and OSA groups in respiratory parameters (Table 3).

Only in the CSA group was there a direct proportional relationship between LVEF (Figure 1A) or SV (Figure 1B) and AHI (*p* = 0.02 and *p* = 0.07). In addition, a direct proportional relationship between LVEF (Figure 1C) or SV (Figure 1D) and TST90 was seen (*p* = 0.06 and *p* = 0.07). In the OSA group, none of the respiratory parameters showed a significant relationship with echocardiographic parameters assessing LV morphology and function.

In the OSA group, the size of the RV showed a positive relationship with AHI (for RVD1 *p* = 0.06), mean A/H (for RVD1 *p* = 0.03, for RVOT proximal *p* = 0.009) (Figure 2A,B), and max A/H (for RVD1 *p* = 0.049, for RVOT proximal *p* = 0.006) (Figure 2C,D). However, the CSA group had no significant correlations of RV parameters with respiratory parameters.

## 4. Discussion

In our study, we evaluated 84 optimally treated patients with HFrEF, in NYHA class of 2.5 ± 0.5, and with LVEF of 25.5 ± 6.8%. Based on type I polysomnography, SDB—mainly CSA—was diagnosed in 90.5% of participants. In the CSA and OSA subgroups, we analyzed the correlations between echocardiographic parameters (i.e., LVEF and SV) with selected polysomnographic parameters (i.e., AHI, mean and maximum A/H duration, and TST90). Respiratory parameters that determine the SDB severity were associated with LV systolic function and SV only in the CSA group, while RV size correlated primarily with the duration of apnea/hypopnea episodes in OSA.

In the study group, we found at least moderate SDB (AHI ≥ 15/h) in 67.9% of patients. Therefore, this study indicates a high SDB prevalence in patients with HFrEF, especially men. Similar observations have been reported before [1,2,3,4,5,32]. In the multicenter SchlarHF registry [33] involving 6876 patients with LVEF ≤ 45% and NYHA ≥ II, SDB of at least moderate severity (AHI ≥ 15/h) was diagnosed in 46% of patients, also showing male predominance (49%). Compared to the SchlarHF registry patients, the patients in our study had lower LVEF (25.5% vs. 34.0%), were slightly older (68 vs. 66 years old), and were recruited from hospitalized patients only, hence the likely higher rate of SDB diagnosis. Oldenburg et al. [2], in a study of 700 outpatients aged 64.5 ± 10.4 with stable HF (NYHA 2.7 ± 0.5, LVEF 28.3 ± 6.8%), reported SDB (AHI > 5/h) in 76%, with a predominance of men. However, the Oldenburg patients had an equal incidence of CSA (40%) and OSA (36%).

In the present study, CSA was highly prevalent (reported in 76.2%, compared to 14.3% of OSA). However, even in the CSA group, obstructive apneas accounted for 40.21%. Nocturnal rostral fluid shifting from the lower extremities toward the throat tissues, causing an increase in neck circumference, is a specific risk factor for OSA in HFrEF [18,19]. In the study group, OMT use and the absence of circulatory decompensation symptoms were the criteria for inclusion; hence, fluid overload’s impact on OSA occurrence in the presented group was limited. Obesity and wide neck circumference are recognized risk factors for OSA [6,34]. CSA patients, who significantly outnumbered OSA patients, had a lower BMI (30.1 ± 5.3 kg/m^2^ vs. 34.3 ± 10.3 kg/m^2^), and average neck circumference (41.6 ± 3.9 cm vs. 42.9 ± 3.7 cm), which also did not favor OSA. Higher recognition of SDB, similar to the one presented in this analysis, was reported by Damy et al. [32] in a group of 384 patients aged 59 ± 13 years with HFrEF, and mean LVEF of 29 ± 9%. In that analysis, SDB with AHI > 5/h was present in 87%, CSA in 62%, and OSA in 26% of participants.

The coexistence of CSA and OSA in the presented group highlights the difficulty in addressing an appropriate treatment of SDB in HF patients. The safety and effectiveness of positive airway pressure (PAP) therapy differ between patients with CSA and OSA [35]. First, the central-to-obstructive-events ratio in the same patient may vary overnight (depending on sleep phase) and in longer follow-up (depending on fluid overload and the presence of pulmonary congestion) [4,23,24]. Second, even with a significant percentage and severity of obstructive episodes, PAP may not be fully effective due to residual central apneas [35]. Third, hypopneas constitute a high percentage of respiratory events in most patients. However, these episodes were often not nominally classified as central or obstructive. Although certain characteristic features make such a division possible [29], it is sometimes time-consuming and error-prone. In our study, we also did not evaluate the types of hypopneas. Unfortunately, failure to divide hypopneas into central and obstructive may hinder accurate and reliable assessment of the predominant pathophysiology of SDB in a given HF patient and the chance for effective and safe PAP therapy. Nevertheless, carefully conducted and interpreted PSGs are needed in HF patients to reach an accurate diagnosis and clear the path for optimal treatment [35].

Although our study showed a trend toward a higher ESS score for the OSA group, (*p* = 0.08, the average ESS score for OSA was 10.1 ± 4.9 points, while for the CSA group it was 7.6 ± 4.4 points), the percentage of participants with pathological daytime sleepiness in the CSA and the OSA groups did not differ significantly. That may indicate a subjective lack of excessive daytime sleepiness in CSA patients. These results correspond with the observations of other researchers [4,33,36,37]. Hastings et al. [36] found an absence of daytime sleepiness in patients with HF with concomitant SDB, and the median ESS scores of 7 (IQR 2–16) and 9 (IQR 2–17) for the patients with and without SDB, respectively. Hastings’ patients with SDB spent more time in bed and were less active during the day than patients without SDB. These data underscore the fact that patients with HF and SDB may underestimate the symptoms of daytime sleepiness. The reduced perception of sleepiness may be due to reduced daytime activity levels with prolonged periods of bed rest [38]. On the other hand, the persistently enhanced sympathetic activity seen in HFrEF, compounded by recurrent apnea, increases arousal and wakefulness and may decrease symptoms of daytime sleepiness [39]. In the Taranto Montemurro et al. study [40], subjective daytime sleepiness decreased with increasing sympathetic activity.

In contrast to the other observations [2,41], male sex did not predispose one to a higher incidence of OSA, which may result from the small size of the OSA group. Patients with severe HFrEF and OSA also did not differ from those with CSA in terms of parameters evaluating CRF, such as NYHA and 6 MWT. However, we observed a trend toward IVS hypertrophy in the OSA group (10.4 ± 2.4 mm vs. 12.0 ± 4.3 mm, *p* = 0.06). IVS thickness of >12 mm was an echocardiographic predictor for OSA in Szymański et al. [42]. It is known that increased afterload caused by refractory hypertension and a repeated increase in LV wall tension caused by inspiration against a closed airway during apnea/hypopnea episodes may stimulate concentric LV hypertrophy in patients with OSA [43].

Moreover, we found differences between the CSA and OSA groups in the relationships of respiratory parameters and LV and RV size or function.

We observed a positive correlation between LVEF or SV, and AHI and TST90 in the CSA group. It is known that CSA occurs when the PCO_2_ falls below the driving threshold of the brain’s respiratory control center, with the length of the cycle of hypocapnia-induced apnea and reflex hyperventilation inversely related to cardiac output [44]. Prolonging the respiratory cycle limits the possible number of respiratory events. Thus, the AHI value is inversely related to the severity of HF assessed as LVEF/SV and is paradoxically lower in patients with more impaired systolic function. However, TST90 is positively correlated with the LVEF or SV, resulting from the more frequent occurrence of respiratory events with desaturation <90%. In a study of 104 patients with HFrEF and CSA–CSR (AHI ≥ 15/h), Wedewardt et al. [45] did not find an association between LVEF impairment and AHI. However, they noted that the respiratory parameters that characterize CSA–CSR (such as respiratory cycle length, apnea length, and ventilation length) are correlated with cardiac dysfunction and could be helpful in cardiac function monitoring. However, we did not assess the parameters of the ventilation phase in our cohort (only apneas and hypopneas). Spiesshoefer et al. also found no correlations between cardiac function and SDB severity when using traditional markers of SDB severity (events/h) but reported that ventilation length increased with worsening LVEF [46].

Interestingly, we found no correlation between LV systolic function and respiratory parameters in patients with dominant OSA, which could be explained by a different pathomechanism of this disease. Nevertheless, analysis of the influence of OSA severity on RV showed that, along with the increase in mean A/H and max A/H, there was a proportional increase in RVOT proximal and RVD1 dimensions. Previous studies have reported an independent effect of OSA on the RV [47,48], which indicates a correlation between AHI and RV dimensions or TAPSE [49,50]. Various mechanisms underlie the changes in RV morphology and function in OSA patients [51]. One possible mechanism is the effect of negative intrathoracic pressure, generated by breathing against obstructed airways, on increased venous return and RV volume overload [52]. Longer episodes of apnea consequently subject the circulatory system to harmful fluctuations in intrathoracic pressure and deterioration of RV function over a long period. Moreover, hypoxia-induced pulmonary arteries’ vasoconstriction, followed by an elevation in pulmonary artery pressure, also turns up RV afterload [53], further deteriorating RV function.

In our cohort of participants aged 68.3 years, AF was more prevalent in the CSA than in the OSA group. Many retrospective clinical studies have concentrated on the association between AF and OSA or nocturnal oxygen desaturation [54,55,56]. However, in the analysis by Tung et al. [57] of 2912 participants from the Sleep Heart Health Study (mean age of 62.8 years), CSA and CSA–CSR with central apnea index of ≥five events per hour were associated with increased risk of developing AF (OR 3.00, and OR 1.83). Conversely, an obstructive apnea index ≥ five events per hour was not a good predictor of AF incidents. Additionally, according to data from the Outcomes of Sleep Disorders Study in Older Men [58], CSA and CSA–CSR predispose to the development of AF, and its impact is even higher in older participants aged ≥ 76 (OR 9.97 and 6.31). Similar findings are reported by Anzai et al. [59] in the Japanese population. Given the unfavorable influence of AF on the prognosis of patients with HFrEF [60], the association of CSA and AF may have severe clinical implications.

### Study Limitations

The study group consisted of patients with significantly reduced LVEF of ischemic or non-ischemic origin and with the presence or absence of intraventricular conduction disorders. That created a very diverse group. In addition, patients had multiple comorbidities that could affect the SDB profile and were not analyzed. A limitation of the statistical analysis was the small group of patients with dominant OSA. Moreover, women accounted for only 20% of patients in the study cohort. When assessing PSG examinations, we counted the episodes of hypopneas without dividing them into central and obstructive. Only in individual cases, when hypopneas predominated while the number of apneas was relatively small, did we consider their type when differentiating CSA and OSA. After completing the study, no long-term follow-up was performed to draw additional conclusions.

## 5. Conclusions

Sleep-disordered breathing is common in patients with systolic heart failure. In HFrEF patients, respiratory parameters determining the severity of SDB are related to LV systolic function and SV only in CSA. In turn, the RV size correlates primarily with the duration of apnea/hypopnea episodes in OSA.

## Figures and Tables

**Figure 1 diagnostics-13-03309-f001:**
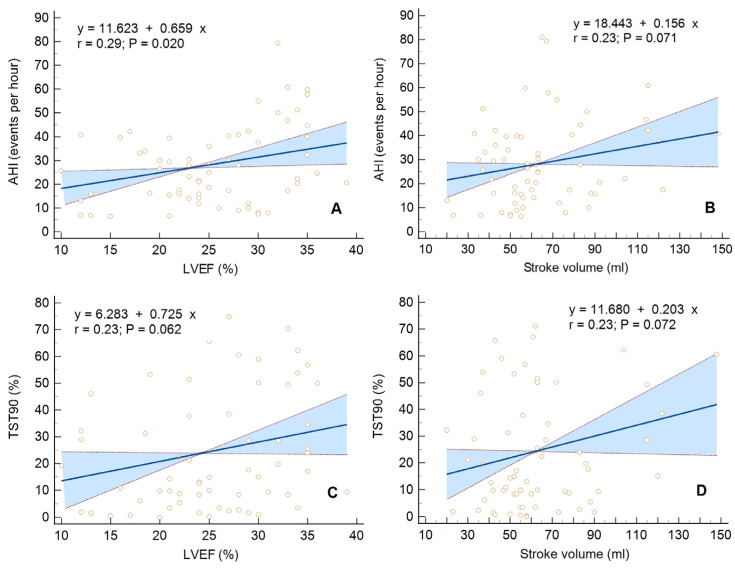
Univariate linear regression with a 95% confidence interval for the following relationships: (**A**) Apnea/hypopnea index (AHI) vs. left ventricular ejection fraction (LVEF), (**B**) AHI vs. left ventricular stroke volume, (**C**) The percentage of total sleep time spent with oxyhemoglobin saturation below 90% (TST90) vs. LVEF, (**D**) TST90 vs. left ventricular stroke volume, in the group of patients with central sleep apnea.

**Figure 2 diagnostics-13-03309-f002:**
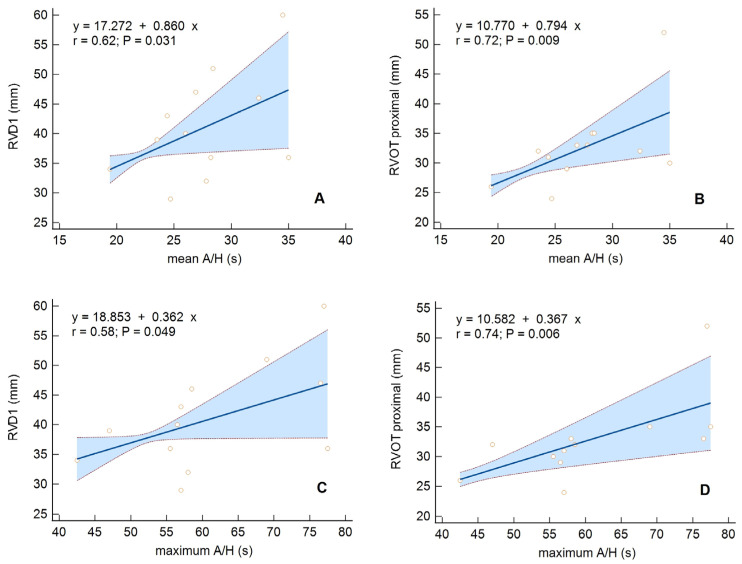
Univariate linear regression with a 95% confidence interval for the following relationships: (**A**) Mean duration of a respiratory event (mean A/H) vs. basal right ventricular linear dimension (RVD1), (**B**) Mean A/H vs. proximal right ventricular outflow diameter (RVOT proximal), (**C**) Maximum duration of a respiratory event (maximum A/H) vs. RVD1, (**D**) Maximum A/H vs. RVOT proximal, in the group of patients with obstructive sleep apnea.

**Table 1 diagnostics-13-03309-t001:** Characteristics of anthropometric parameters, comorbidities, and pharmacological treatment in surveyed groups with central and obstructive sleep apnea.

	All Cases	CSA Group	OSA Group	*p*-Value
	Anthropometric parameters			
No (%)	76	64 (76,2%)	12 (14,3%)	
Sex (male), no (%)	64	54 (84.4%)	10 (83.3%)	0.93
Sex (female), no (%)	12	10 (15.6%)	2 (16.7%)	0.93
Age (years), X ± SD	68.0 ± 8.7	67.9 ± 8.8	68.3 ± 9	0.91
BMI (kg/m^2^), X ± SD	30.7 ± 6.4	30.1 ± 5.3	34.3 ± 10.3	0.001
Neck circumference (cm), X ± SD	41.9 ± 3.8	41.7 ± 3.8	42.9 ± 3.7	0.98
	Comorbidities			
CAD with a history of MI, no (%)	53 (69.7%)	45 (70.3%)	8 (66.7%)	0.06
Hypertension, no (%)	60 (78.9%)	50 (78.1%)	10 (83.3%)	0.90
Diabetes, no (%)	40 (52.6%)	33 (51.6%)	7 (58.3%)	0.09
Atrial fibrillation, no (%)	27 (35.5%)	26 (40.6%)	1 (8.3%)	0.03
CKD, no (%)	16 (21.1%)	13 (20.3%)	3 (25.0%)	0.30
PAD, no (%)	8 (9.5%)	4 (6.2%)	3 (25.0%)	0.10
Hypothyroidism, no (%)	6 (7.9%)	6 (9.4%)	0 (0.0%)	0.50
Stroke, no (%)	4 (5.3%)	4 (6.2%)	0 (0.0%)	0.50
CAS, no (%)	5 (6.6%)	5 (7.8%)	0 (0.0%)	0.50
	Pharmacological treatment			
ACE-I, no (%)	60 (78.9%)	49 (76.6%)	11 (91.7%)	0.24
ARB, no (%)	2 (2.6%)	2 (3.1%)	0 (0.0%)	0.54
MRA, no (%)	67 (88.2%)	56 (87.5%)	11 (91.7%)	0.68
Loop diuretics, no (%)	66 (86.8%)	55 (85.9%)	11 (91.7%)	0.59
Beta-blockers, no (%)	75 (98.7%)	63 (98.4%)	12 (100.0%)	0,67
Digitalis, no (%)	6 (7.9%)	6 (9.4%)	0 (0.0%)	0.27
Ivabradine, no (%)	15 (19.7%)	14 (21.9%)	1 (8.3%)	0.28

The results are given for all cases and each of the subgroups. ACE-I: angiotensin-converting enzyme inhibitors; ARB: angiotensin-receptor blockers; BMI: body mass index; CAD: coronary artery disease; cm: centimeter; CAS: carotid artery stenosis; CKD: chronic kidney disease with an estimated glomerular filtration rate of <60 mL/min/1.73 m^2^; CSA: central sleep apnea; MI: myocardial infarction; no: number; MRA: mineralocorticoid-receptor antagonists; OSA: obstructive sleep apnea; PAD: peripheral arterial disease; SD: standard deviation; X: arithmetic mean.

**Table 2 diagnostics-13-03309-t002:** Characteristics of cardiorespiratory fitness and echocardiographic parameters in surveyed groups with central and obstructive sleep apnea.

		All Cases	CSA Group	OSA Group	*p*-Value
		Cardiorespiratory fitness parameters			
No (%)		76	64 (76.2%)	12 (14.3%)	
HF etiology (ICM), no (%)		54 (71.1%)	46 (71.9%)	8 (66.7%)	0.71
HF duration [months], X ± SD		74.8 ± 82.4	81.5 ± 86,8	39.2 ± 37.8	0.10
NYHA class, X ± SD		2.4 ± 0.5	2.4 ± 0.56	2.5 ± 0.52	0.66
6 MWT [m], X ± SD		282.2 ± 105	282.1 ± 110.3	281.7 ± 76.9	0.99
		Echocardiographic parameters			
LVEDD [mm], X ± SD		67.80 ± 7.91	67.92 ± 8.39	67.17	0.048
LVESD [mm], X ± SD		57.22 ± 9.31	57.75 ± 9.68	54.42 ± 6.64	0.17
IVS [mm], X ± SD		10.6 ± 2.8	10.4 ± 2.4	12.0 ± 4.3	0.06
PW [mm], X ± SD		9.59 ± 1.72	9.55 ± 1.66	9.75 ± 2.09	0.25
LVEF [%], X ± SD		25.4 ± 6.8	25.2 ± 7.2	25.9 ± 6	0.76
SV [mL], X ± SD		64.5 ± 24.4	63.8 ± 25.8	68.7 ± 14.7	0.53
LVEDVI [mL/m^2^], X ± SD		129.1 ± 39.9	129.5 ± 40.6	128.3 ± 39.2	0.93
LVESVI [mL/m^2^], X ± SD		96.6 ± 34.5	97.2 ± 34.9	95.3 ± 36.4	0.87
LAA [cm^2^], X ± SD		27.64 ± 6.60	27.92 ± 6.68	26.17 ± 6.24	0.86
MR	trace/mild, no (%)	39 (51.3%)	32 (50.0%)	7 (58.3%)	0.60
	moderate/severe, no (%)	37 (48.7%)	32 (50.0%)	5 (41.7%)	
RVOTproximal [mm], X ± SD		34.2 ± 5.8	34.5 ± 5.6	32.7 ± 6.9	0.33
RVD1 [mm], X ± SD		38.8 ± 6.9	38.5 ± 6.5	41.1 ± 8.8	0.24
TAPSE [mm], X ± SD		18.3 ± 4.7	17.9 ± 4.3	20.6 ± 6.2	0.07
RAA [cm^2^], X ± SD		20.13 ± 6.10	20.38 ± 6.34	18.83 ± 4.61	0.25
TR	trace/mild, no (%)	53 (71.6%)	45 (71.4%)	8 (72.7%)	0.93
	moderate/severe, no (%)	21 (28.4%)	18 (28.6%)	3 (27.3%)	
Peak TR velocity [m/s], X ± SD		2.77 ± 0.80	2.76 ± 0.82	2.85 ± 0.72	0.88
TRPG [mm Hg], X ± SD		31.21 ± 15.28	30.76 ± 15.30	34.56 ± 16.40	0.70
RVSP [mm Hg], X ± SD		40.4 ± 17.7	40.1 ± 17.8	42.8 ± 19.6	0.76

The results are given for all cases and each of the subgroups. CSA: central sleep apnea; HF: heart failure; ICM: ischemic cardiomyopathy; IVS: interventricular septum; LAA: left atrial area; LVEDD: end-diastolic dimension; LVEDVI: left ventricular end-diastolic volume index; LVEF: left ventricular ejection fraction; SV: stroke volume; LVESD: end-systolic dimension; LVESVI: left ventricular end-systolic volume index; no: number; MR: mitral valve regurgitation; NYHA: The New York Heart Association functional classification; OSA: obstructive sleep apnea; PW: posterior wall of the left ventricle; RAA: right atrial area; RVD1: basal right ventricle linear dimension; RVOT proximal: proximal right ventricle outflow diameter; RVSP: right ventricular systolic pressure; SD: standard deviation; TAPSE: tricuspid annular plane systolic excursion; TR: tricuspid valve regurgitation; TRPG: peak systolic tricuspid pressure gradient; X: arithmetic mean; 6 MWT: 6 min walk test.

**Table 3 diagnostics-13-03309-t003:** Characteristics of polysomnographic parameters in surveyed groups with central and obstructive sleep apnea.

	All Cases	CSA Group	OSA Group	*p*-Value
	Polysomnographic Parameters			
No (%)	76	64 (76.2%)	12 (14.3%)	
ESS (points), X ± SD	8.0 ± 4.5	7.6 ± 4.4	10.1 ± 5.0	0.08
Participants with ESS score >10 pts, no (%)	20 (26.3%)	16 (25.0%)	4 (33.3%)	0.55
Percentage of central apneas, M (IQR)	52.41 (20.72–78.17)	57.33 (29.29–84.09)	20.20 (1.49–42.92)	0.003
Percentage of obstructive apneas, M (IQR)	46.60 (20.81–76.59)	40.21 (13.83–69.79)	79.80 (57.09–98.52)	0.002
AHI [events/h], X ± SD	27.9 ± 17.0	28.4 ± 17.3	25.8 ± 15.8	0.77
mean A/H [s], X ± SD	28.3 ± 4.5	28.7 ± 4.4	27.6 ± 4.6	0.44
max A/H [s], X ± SD	59.2 ± 10.6	60.3 ± 9.1	61.0 ± 11.6	0.83
mean SpO_2_ [%], X ± SD	91.1 ± 2.0	91.1 ± 2.1	91.1 ± 2.1	0.99
min SpO_2_ [%], X ± SD	78.6 ± 7.9	78.6 ± 7.8	77.1 ± 8.1	0.54
TST 90 [%], X ± SD	23.5 ± 21.4	24.7 ± 22.5	21.2 ± 18.0	0.62

The results are given for all cases and each of the subgroups. AHI: apnea–hypopnea index; A/H: apneic or hypopneic events; CSA: central sleep apnea; ESS: Epworth Sleepiness Scale; max: maximal; mean: average; no: number; IQR: interquartile range; M: median; OSA: obstructive sleep apnea; SD: standard deviation; SpO_2_: oxygen saturation estimated by pulse oximetry, TST 90: the percentage of total sleep time spent with oxyhemoglobin saturation below 90%; X: arithmetic mean.

## Data Availability

The data presented in this study are available on request from the Department of Electrocardiology and Heart Failure, Medical University of Silesia in Katowice (Poland). The data are not publicly available due to privacy restrictions.

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
