# Peer review of "Relationships between Heart Chamber Morphology or Function and Respiratory Parameters in Patients with HFrEF and Various Types of Sleep-Disordered Breathing"

_diagnostics, 2023, doi:10.3390/diagnostics13213309_

Round 1
Reviewer 1 Report
The article is very concise and to the point. It was a pleasure to read it!
Author Response
We would like to thank you for accepting the manuscript.
Reviewer 2 Report
Dear authors, your article shows he results of a transvertial study of the reationship of heart morphology in SDB patients. The methods are in accordance to your objectives. I suggest to include the description of all data obtained by echocardiography (RVD, RVOT, PAPSE IVS etc).
PSG analysis please make sure there is no typing mistake, for hypopnea definition (30% volume reduction...would it be 50%?).
The results are well presented. I suggest to include in table 1 data for female patients (distribution for CSA and OSA - similar to men?).
In table 3: I suggest to show also the index of obstructive apneas and central apneas (oAI and cAI) for the patients as many?all? have both events, even if at the end you diagnose or CSA or OSA. It would be interesting to include the importance of this " composed disease" in the discussion with implication on treatment.
ESS as an absolute number is important? How many scored "sleepy" (ESS>10) in each group?
Figure 3: the text reports "...in a subgroup of OSA patientes". ..It was not clear to me if you included all OSA patients for the analysis or just some (and which ones)....necessary to improve the writing to make it more clear.
You found an association of artial fibrilation and CSA, comment.
English writing is good, some parts should be revised as the sentence structures are not ok.
Author Response
I suggest to include the description of all data obtained by echocardiography (RVD, RVOT, PAPSE IVS etc).
Reply: Thank you for this comment.
In the "Echocardiographic Measurements" section, we supplemented the description of echocardiographic parameters crucial for the study. We also updated Table 2 with additional echocardiographic parameters.
PSG analysis please make sure there is no typing mistake, for hypopnea definition (30% volume reduction...would it be 50%?).
Reply: Thank you for this remark.
We corrected a typing error.
The results are well presented. I suggest to include in table 1 data for female patients (distribution for CSA and OSA - similar to men?).
Reply: Thank you for this remark.
We updated Table 1 with data describing the distribution of CSA and OSA participants in females. The percentage distribution of patients with CSA and OSA did not differ in the groups of men and women (p=0.93). However, the number of women in the OSA group was low (2 patients).
In table 3: I suggest to show also the index of obstructive apneas and central apneas (oAI and cAI) for the patients as many?all? have both events, even if at the end you diagnose or CSA or OSA. It would be interesting to include the importance of this " composed disease" in the discussion with implication on treatment.
Reply: Thank you for this valuable remark.
We did not calculate the index of specific types of respiratory events–central apneas, obstructive apneas and hypopneas–in our study. However, we calculated the percentage of central and obstructive apneas in the CSA and the OSA groups. We added the results to Table 3.
Most patients had both types of apnea–central and obstructive. Only in 5 participants, the rate of central apneas was 100%, and in 4 participants, the rate of obstructive apneas was 100% (added to the Results section). However, in most patients, a high percentage of respiratory events were hypopnea episodes. These episodes were not nominally classified as central or obstructive. Although some characteristic features make such a division possible, it is sometimes time-consuming and error-prone. Therefore, in medical practice and most scientific research, the number of hypopneas is counted together. In our study, we also did not evaluate the types of hypopneas. However, in individual cases, when hypopneas predominated, while the number of apneas was relatively small, we considered their type when differentiating CSA from OSA. Unfortunately, failure to divide hypopneas into central and obstructive may result in missing a significant number of respiratory events if only the central/obstructive apnea index is used, e.g., to assess the chance for successful PAP therapy in patients with HF and SDB.
Perhaps the sort out all respiratory episodes (into central or obstructive) in patients with HF should be considered. Additionally, it may be appropriate to separately assess the severity (number, duration, desaturations) of central vs. obstructive respiratory events (apneas and hypopneas in total) to verify the predominant pathophysiological problem and the safety of PAP therapy in a given HF patient. However, despite a significant severity of obstructive episodes, PAP may not be fully effective due to residual central apneas. Unfortunately, in our cardiology center we do not have the possibility of PAP treatment, and we cannot verify this assumption. Hopefully, another researcher will find this idea interesting.
We covered this topic in the Discussion section. We described the lack of division of hypopneas into central and obstructive in our study group as the limitation of the study.
ESS as an absolute number is important? How many scored "sleepy" (ESS>10) in each group?
Reply: Thank you for this question.
A total of 20 (26.3%) participants reported increased daytime sleepiness with ESS score >10 pts, including 16 (25.0%) with CSA and 4 (33.3%) with OSA. No statistically significant difference was obtained (p=0.55), which, despite the small OSA group, may confirm the lack of pathological daytime sleepiness in patients with cardiovascular diseases described in the literature. Data are supplemented in Table 3.
Figure 3: the text reports "...in a subgroup of OSA patientes". ..It was not clear to me if you included all OSA patients for the analysis or just some (and which ones)....necessary to improve the writing to make it more clear.
Reply: Thank you for this remark.
We analyzed all patients with CSA and all patients with OSA without exclusions. We corrected misleading figure descriptions.
You found an association of artial fibrilation and CSA, comment.
Reply: Thank you for this comment.
In our cohort, AF was more prevalent in the CSA than in the OSA group. Many retrospective clinical studies have concentrated on the association between AF and OSA or nocturnal oxygen desaturation. However, in the analysis of data from the Sleep Heart Health Study and from the Outcomes of Sleep Disorders Study in Older Men, CSA and the CSA-CSR with central apnea index of ≥ five events per hour (but not obstructive apnea index) were associated with increased risk of developing AF. Given the unfavorable influence of AF on the prognosis of patients with HFrEF, the association of CSA and AF may have severe clinical implications.
We covered this topic in the Discussion section.
Comments on the Quality of English Language
English writing is good, some parts should be revised as the sentence structures are not ok.
Reply: Thank you for this comment.
All text was proofread by a native speaker. However, when revising the manuscript, we made efforts to improve the sentence structures.
Reviewer 3 Report
Dear authors,
I have studied with great interest the manuscript “Relationships between heart chamber morphology or function and respiratory parameters in patients with HFrEF and various types of sleep-disordered breathing”.
The main question addressed by the research was to evaluate the relationships between heart chamber size or function and respiratory parameters in patients with HFrEF and various types of SDB. The study highlighted that in HFrEF patients, SDB severity is related to LV systolic function and SV only in CSA, whereas RV size correlates primarily with apnea/hypopnea episode duration in OSA.
The manuscript is clearly exposed and well written. The figure and tables correspond to the description in the text, are well designed and reflect important information. The references are appropriate. The topic is original.
I have some comments to improve the quality of the presentation.
1. In the section “Echocardiographic Measurements” the information about other parameters measurement should be presented.
2. The NYHA classes and prescribed medical therapy should be included in patient’s characteristics.
Author Response
- In the section “Echocardiographic Measurements” the information about other parameters measurement should be presented.
Reply: Thank you for this comment.
In the "Echocardiographic Measurements" section, we supplemented the description of echocardiographic parameters crucial for the study. We also updated Table 2 with additional echocardiographic parameters.
- The NYHA classes and prescribed medical therapy should be included in patient’s characteristics.
Reply: Thank you for this comment.
Data on pharmacological treatment was added to Table 1. The NYHA classes have already been included in Table 2., titled "Characteristic of cardiorespiratory fitness and echocardiographic parameters in surveyed groups with central and obstructive sleep apnea." We hope that this data arrangement is legible.
Round 2
Reviewer 3 Report
The authors improved the paper